# Facile Hydrothermal Synthesis of Ag/Fe_3_O_4_/Cellulose Nanocomposite as Highly Active Catalyst for 4-Nitrophenol and Organic Dye Reduction

**DOI:** 10.3390/polym15163373

**Published:** 2023-08-11

**Authors:** An Nang Vu, Hoa Ngoc Thi Le, Thang Bach Phan, Hieu Van Le

**Affiliations:** 1Faculty of Materials Science and Technology, University of Science, VNU-HCM, Ho Chi Minh City 700000, Vietnam; vnan@hcmus.edu.vn (A.N.V.); ltnhoa@hcmus.edu.vn (H.N.T.L.); 2Vietnam National University Ho Chi Minh City, Ho Chi Minh City 700000, Vietnam; pbthang@inomar.edu.vn; 3Laboratory of Multifunctional Materials, University of Science, VNU-HCM, Ho Chi Minh City 700000, Vietnam; 4Center for Innovative Materials and Architectures, VNU-HCM, Ho Chi Minh City 700000, Vietnam

**Keywords:** catalytic dye reduction, cellulose nanocrystals, magnetic nanocomposite, 4-nitrophenol, wastewater treatment

## Abstract

Novel effluent treatment solutions for dangerous organic pollutants are crucial worldwide. In recent years, chemical reduction using noble metal-based nanocatalysts and NaBH_4_, a reducing agent, has become common practice for eliminating organic contaminants from aquatic environments. We suggest a straightforward approach to synthesizing magnetic cellulose nanocrystals (CNCs) modified with magnetite (Fe_3_O_4_) and silver nanoparticles (Ag NPs) as a catalyst for organic contamination removal. Significantly, the CNC surface was decorated with Ag NPs without using any reducing agents or stabilizers. PXRD, FE-SEM, TEM, EDX, VSM, BET, and zeta potential tests characterized the Ag/Fe_3_O_4_/CNC nanocomposite. The nanocomposite’s catalytic activity was tested by eliminating 4-nitrophenol (4-NP) and the organic dyes methylene blue (MB) and methyl orange (MO) in an aqueous solution at 25 °C. The Ag/Fe_3_O_4_/CNC nanocomposite reduced 4-NP and decolored these hazardous organic dyes in a short time (2 to 5 min) using a tiny amount of catalyst (2.5 mg for 4-NP and 15 mg for MO and MB). The magnetic catalyst was removed and reused three times without losing catalytic activity. This work shows that the Ag/Fe_3_O_4_/CNC nanocomposite can chemically reduce harmful pollutants in effluent for environmental applications.

## 1. Introduction

Environmental pollution has become a global and grave threat to human health despite the exponential development of modern industries over the past few decades. In recent years, the contamination of water bodies by releasing untreated water containing inorganic and organic species has garnered significant attention among the various types of environmental pollution [1,2,3]. Notably, it is believed that the uncontrolled discharge of azo dyes, widely used in the leather, fabric, plastic, personal care products, ink, paper, and food industries, causes significant damage to aquatic ecosystems [4,5]. Due to their photo-, thermal-, and biodegradation-resistance, organic dyes can endure for an extended period in the environment. The existence and stability of these oxygen-sequestering species in water systems have been reported to diminish light penetration and hinder photosynthesis in aquatic vegetation [6]. Before they can be discharged safely into the natural environment, industrial effluents carrying harmful organic dyes must be degraded and decolored.

In addition to organic azo pigments, the United States Environmental Protection Agency (U.S. EPA) considers phenolic compounds, such as the well-known nitrophenol derivatives, to be priority pollutants that directly affect the environment and human health [7]. Known refractory 4-nitrophenol (4-NP) sources in water bodies include petrochemical, pesticide, pharmaceutical, preservative, explosive, pigment, and wood industries [8,9]. Before releasing them into the environment, developing efficient techniques for extracting them from industrial effluent is critical. The catalytic conversion of 4-NP to its amino counterpart, 4-aminophenol (4-AP), has generated significant interest in environmental cleanup. The essential roles amino acids perform in pharmaceutical and photographic applications are well known [10]. 4-AP is a critical intermediate in synthesizing numerous antipyretic and analgesic pharmaceuticals, while its strong reducing properties are utilized in photographic development. Therefore, developing high-performance materials for pigment degradation and 4-NP reduction can significantly reduce environmental pollution. In recent years, nano-sized metals with controlled morphology and texture have been favored over their bulk counterparts because of their enhanced catalytic, magnetic, electronic, and optical properties [11,12,13]. Notably, noble metal nanoparticles exhibit dense packing, unconstrained electrons in the valence band, and a high surface-to-volume ratio [14]. Due to their desirable physical and chemical properties [15,16], using inexpensive silver nanoparticles (Ag NPs) in catalysis, sensors, cosmetics, and the disinfection of medical devices is becoming increasingly investigated. Controlled growth, particle size, morphology, and long-term stability play significant roles in the performance activity of Ag NPs.

Nevertheless, metal nanoparticles with desirable morphological properties as catalysts have disadvantages, such as (i) a tendency to aggregate, resulting in decreased efficiency due to their more incredible surface energy and (ii) difficult recovery from the medium of reaction for reuse due to their nanoscale size [17,18]. To address these problems, scientists have devised new ways to fix metal nanoparticles on solid supports to make hybrid nanomaterials. Interestingly, the direct immobilization of Ag NPs on other active materials, such as Fe_3_O_4_, makes them ideally suited for use as a catalyst; unfortunately, the repulsive forces between them must be surmounted to ensure their stability. Introducing a buffer (organic) layer between the two active materials can alleviate the issue of repulsive forces and bring about strong synergy in the hybrid structure for efficient catalytic activity [19,20,21,22].

In recent years, biomass has attracted concern regarding the design of nanostructure catalysts based on a porous structure and numerous active sites for fixing metal nanoparticles [23,24,25,26]. Specifically, cellulose is the overall biomass most extensively studied to produce stable metal nanoparticles [27,28,29,30,31]. By functionalizing polydopamine-containing porous cellulose acetate microspheres, Li et al. generated substrates to synthesize Ag-Fe_3_O_4_ nanoparticles [27]. The Fe–Cu alloy nanocatalyst is immobilized in cellulose microcrystals. These compounds are coupled with NaBH_4_ in water to convert nitroarene to arylamine. The reaction takes 5 to 14 min and is effective [32]. Moreover, cellulose can be transformed into various polymorphs using an environmentally friendly mixture of NaOH and urea [33,34,35], and through hydrolysis, cellulose nanocrystals (CNCs) with high physical properties can be obtained.

In this research, a simple and environmentally friendly hydrothermal method for preparing Ag/Fe_3_O_4_/CNC nanocomposites was developed. The Ag/Fe_3_O_4_/CNC nanocomposite’s phase structure, morphology, magnetic property, and thermal stability were investigated in detail. In addition, catalytic reduction experiments were conducted, and the results demonstrated that the as-prepared Ag/Fe_3_O_4_/CNC nanocomposites had a high catalytic performance for the reduction of 4-NP, MO, and MB in the presence of sodium borohydride (NaBH_4_) as an electron donor. Compared to previous works, the current work possesses many advantages. First, neutral deionized water was used as the system’s solvent. Second, the green and inexpensive synthetic route requires no chemical-reducing agents. In addition, CNCs were produced by acid-hydrolysis cellulose, which was isolated from the by-product fibers of coconut husk. It is one of the most effective methods for creating economic materials from agricultural by-products, which are nearly abundant in Vietnam.

## 2. Materials and Methods

### 2.1. Materials

Coconut husk fiber was collected from Mo Cay Nam district, Ben Tre province. After the coconuts were picked, the insides were separated from the shells. Then, the coconuts were crushed and dried and the weakened fiber was removed from the coconut shells. Finally, the coconuts were split into coconut fiber. Coconut fiber is 10–20 cm long, yellowish-brown in color, and twisted together, which was crushed into powder. Formic acid (HCOOH, 90%), hydroperoxide (H_2_O_2_, 30%), sodium hydroxide (NaOH, 96%), hydrochloric acid (HCl, 37%), urea ((NH_2_)_2_CO, ≥99%), ethylene glycol (HOCH_2_CH_2_OH, ≥99%), ethanol (C_2_H_5_OH, 99,5%), and two precursors of Fe and Ag (iron(III) chloride hexahydrate (FeCl_3_.6H_2_O, 97%) and silver nitrate (AgNO_3_, 99%)) were purchased from XiLong, China. MO (C_14_H_14_N_3_NaO_3_S, ≥95%), MB (C_16_H_18_N_3_SCl, ≥95%), sodium borohydride (NaBH_4_, ≥99%), and 4-NP (≥99%) were purchased from Sigma-Aldrich. All the chemicals and reagents were used as received without further purification.

### 2.2. Isolation of Cellulose from Coconut Fiber and Hydrolysis to Generate CNCs

The isolation of cellulose from coconut fiber using the formic/peroxyformic acid process was conducted as described in our previous work [36]. There are three major steps to isolate cellulose: treatment with formic acid (HCOOH), treatment with peroxyformic acid (PFA-HCOOH + H_2_O_2_ + H_2_O mixture), and bleach with NaOH and H_2_O_2_ solution. First, coconut fiber powder was stirred with distilled water at 90 °C for 2 h following a ratio of 1:20 (coconut fiber weight:volume of water). Next, the sample was filtered, washed in distilled water, and dried at 60 °C. Then, the coconut fiber was stirred with a reflux condenser in acid HCOOH 90% under the coconut fiber weight:HCOOH volume of 1:10 at 90 °C for 2 h. After HCOOH treatment, the mixture was filtered and washed with distilled water to remove the excess acid. The sample was dried at 60 °C. The coconut fiber was continuously treated with a PFA mixture (90% HCOOH 90%, 4% H_2_O_2_ 30%, and 6% distilled water) under a 1:20 ratio (fiber weight:PFA volume). The system was mechanically stirred with reflux at 100 °C for 2 h. Finally, the fiber was bleached to achieve pure cellulose. Precisely, 6.0 g of PFA-treated fiber was placed in a 500 mL three-neck round bottom flask, followed by a solution of 180 mL distilled water and 12 mL NaOH 1M, and the system was heated to 70 °C. After that, 16 mL H_2_O_2_ 30% was slowly poured into the system (all H_2_O_2_ was poured within 15 min). The system was kept at a stable heat at 80 °C. The mixture was then filtered and washed in distilled water. The sample obtained was cellulose, which was dried at 80 °C.

The cellulose was hydrolyzed with hydrochloric acid (HCl, 6M) under a 1:25 (cellulose weight:HCl volume) ratio. The system was stirred at a stable temperature of 90 °C for 3 h. When the reaction finished, the mixture was put into a beaker that contained 1000 mL distilled water and the obtained suspension. The suspension was deposited, the water was changed a few times, and the sample was centrifuged at a rate of 6000 rpm for 10 min and then dried at 80 °C. The final sample obtained was CNCs.

### 2.3. Preparation of Fe_3_O_4_ Nanoparticles by Solvothermal Method

The manufacture of Fe_3_O_4_ nanoparticles using the solvothermal technique is depicted in Figure 1. First, 42.0 mL of ethylene glycol and 1.5 mL of distilled water were combined with 0.43 g FeCl_3_·6H_2_O and 0.90 g (NH_2_)_2_CO to generate an orange solution. This mixture was then ultrasonically processed for 30 min. The combined solution was then put into a Teflon-lined stainless-steel autoclave, sealed for six hours, and heated to 220 °C before being cooled to ambient temperature. After obtaining black precipitation of Fe_3_O_4_, the sample was centrifuged five to six times with ethanol. The Fe_3_O_4_ nanoparticles were made after washing and vacuum drying at 60 °C for 9 h.

### 2.4. Preparation of Ag/Fe_3_O_4_/CNC Nanocomposite by Hydrothermal Method 

The experimental procedure involved subjecting a mixture comprising 10 mL of CNCs (10 mg·mL^−1^), 10 mL of Fe_3_O_4_ (2 mg·mL^−1^), and 10 mL of AgNO_3_ (17 mg·mL^−1^) to ultrasonic treatment for 30 min. Subsequently, the mixture was transferred into a Teflon-lined stainless-steel autoclave and subjected to hydrothermal treatment at a temperature of 80 °C for 3, 4, and 5 h. When the hydrothermal process finished, a suspension of Ag/Fe_3_O_4_/CNC nanocomposite was obtained and then centrifuged and washed with DI water several times and ethanol three times before drying at 70 °C for 3 h. The obtained products were labeled FAC3, FAC4, and FAC5, corresponding to the hydrothermal times. In addition, for comparative purposes, Ag/Fe_3_O_4_ (FA) samples were synthesized without CNCs at three separate times: 3 h, 4 h, and 5 h under the same conditions. Like the FAC materials, the FA samples were labeled FA3, FA4, and FA5, corresponding to the hydrothermal times.

### 2.5. Characterization

The crystal structure was characterized by powder X-ray diffraction (PXRD), which was implemented on a Bruker D2 Phaser PXRD (Berlin, Germany) instrument with Cu *Kα* (target) radiation (*λ* = 1.5418 Å) at a scan rate (2*θ*) of 0.02° min^−1^ and a scan range of 10° to 80°. The samples were ground into a fine powder and placed into a groove on a glass slide. After being compacted, the slide with the powder was used for PXRD experiments. The morphology was characterized using a field emission scanning electron microscope (FE-SEM, S4800, Hitachi, Japan) equipped with an energy-dispersive X-ray (EDX) spectrometer for elemental analysis using accelerating voltages of 5 kV and 10 kV. A sample patch was adhered to a specimen stage by conductive adhesive tapes. Before observation, the sample was sputtered with gold for electrical conduction. Transmission electron microscopy (TEM) was performed with a JEM-1400 F (JEOL Ltd., Akishima, Japan) with a field emission gun operating at 100 kV. The sample was suspended in ethanol and was prepared by being drop-cast onto a carbon-coated 200-mesh copper grid and subsequently dried at room temperature. The magnetic properties of the obtained materials were measured with a vibrating sample magnetometer (VSM, LakeShore 7073, Westerville, OH, USA) at 25 °C, and the hysteresis loop was measured in a magnetic field from −12,000 to +12,000 Oe. Thermogravimetric (TG) analysis was performed with a synchronous thermal analyzer (SDTQ600, New Castle, DE, USA) under a nitrogen atmosphere from room temperature to 800 °C at a heating rate of 10 °C·min^−1^.

The specific surface areas of the FA5 and FAC5 were measured using a Quantachrome Nova 2200 (Boynton Beach, FL, USA) BET analyzer and the Brunauer–Emmett–Teller (BET) method based on low-temperature N_2_ adsorption–desorption. The Barrett–Joyner–Halanda (BJH) method was used to calculate the pore size distributions from the adsorption isotherms. The diluted FAC5 suspension was sonicated before DLS analysis. The material was placed in the cuvette to be measured by a zeta analyzer (Nano ZS90, Zetasizer, Malvern, UK).

### 2.6. Catalytic Study

#### 2.6.1. Catalytic Reduction of 4-NP

The conversion of 4-NP to 4-AP served as a model for evaluating the catalytic activity of the FAC5 nanocomposite. In a typical process, 2.5 mg of FAC5 catalyst was added to a 25 mL aqueous 4-NP (1 × 10^−4^ M) while the solution was agitated at 25 °C. Next, a 2.5 mL aqueous solution of NaBH_4_ (1 × 10^−4^ M) was slowly added to the previously described reaction mixture. A sudden change from light yellow to a deeper yellow was observed. Changes in the absorbance at 400 nm for 4-NP and 300 nm for the newly created 4-AP were used to monitor the progression of the -NO_2_ conversion at predetermined intervals. Within 10 min of reaction time, the pigment changed from rich yellow to colorless.

#### 2.6.2. Catalytic Reduction of MO and MB Dyes

In this study, MO and MB were chosen as models for reduction and degradation by FAC5 in the presence of NaBH_4_. In a typical decomposition procedure, 15 mg of FAC5 was added to 30 mL of an aqueous dye solution with a concentration of 1 × 10^−4^ M. The reaction was stirred at 25 °C while adding 1 mL of freshly prepared NaBH_4_ solution (1 × 10^−1^ M). The supernatant’s catalytic activity was determined using UV–vis absorption spectra (*λ*_max_: 464 nm for MO; *λ*_max_: 662 nm for MB) at predetermined time intervals.

## 3. Results and Discussion

### 3.1. Characterization

The PXRD patterns depicted the crystal structure of the Fe_3_O_4_ and Fe_3_O_4_/Ag NPs in Figure 2a. The diffraction peaks at 2*θ* values of 30.1, 35.5, 43.1, 56.9, and 62.6°, which were assigned to the (220), (311), (400), (511), and (440) planes of the face-centered cubic structure of Fe_3_O_4_ (JCPDS card no. 79-0418), confirm the synthesis of Fe_3_O_4_ NPs [37,38]. The peaks of the Fe_3_O_4_/Ag NPs prepared at three different hydrothermal times can be observed at 38.2, 44.6, 64.5°, and 77.5° corresponding to the reflections of the (111), (200), (220), and (311) crystal planes of Ag (JCPDS card no. 87-0720), indicating the face-centered cubic structure of the Ag NPs [39].

Figure 2b depicts the PXRD patterns of the CNCs, Fe_3_O_4_, and three FAC materials. A comparison of the PXRD patterns of the three FAC samples reveals that the characteristic Fe_3_O_4_ and CNC diffraction peaks at the 2*θ* positions are nearly unchanged. However, in the PXRD diffractograms of FAC3 and FAC4, the intensity of the remaining characteristic peaks is comparatively low, except for the silver (111) lattice peak at 2*θ* = 38.2°. In addition, there are impressive peaks at positions 2*θ* = 27.7° and 32.1°; the intensity of these peaks diminished gradually from the 3 h hydrothermal sample to the 4 h hydrothermal sample and disappeared by the 5 h hydrothermal sample. More time may have been required for the CNCs to convert Ag^+^ ions to Ag, resulting in the appearance of these unfamiliar peaks. The reduction reaction had yet to occur fully. When the reaction time is increased to 5 h, the CNCs have sufficient time to convert all the Ag^+^ ions to Ag, resulting in the disappearance of these peaks and an increase in the intensity of the diffraction peaks that characterize the Ag crystal structure.

The FE-SEM image of Fe_3_O_4_ (Figure 3a) reveals that the solvothermal-fabricated Fe_3_O_4_ has a spherical shape with an average diameter of approximately 300 nm. Figure 3b depicts the EDX spectrum of Fe_3_O_4_, which confirms this material has high purity when only Fe and O elements are present, with no other impurities. At three different time intervals, the hydrothermal treatment of Fe_3_O_4_ in the AgNO_3_ solution produced FA materials with remarkably similar morphology. In contrast to Fe_3_O_4_, the formation and attachment of Ag to the surface of Fe_3_O_4_ during the hydrothermal process gives the FA spherical particles an uneven surface (Figure 4). The FAC material indicates that Fe_3_O_4_, Ag, or Ag/Fe_3_O_4_ particles form on the CNC surface in the presence of CNCs. The FE-SEM imaging results indicate that the average particle size in the hydrothermal sample after 3 h is approximately 180 nm; after 4 h, is approximately 150 nm; and after 5 h, is approximately 70–100 nm (Figure 4). When the hydrothermal durations are between 3 and 4 h, the nanoparticles are unevenly distributed and tend to clump together. When the hydrothermal time is sufficiently extended (5 h), the particle density becomes more remarkable than that of the two preceding samples. The 5 h nanoparticles disperse uniformly on the surface of crystalline cellulose and are less susceptible to agglomeration. As the hydrothermal duration increases, the particle density on the CNC surface increases, the particle size decreases, and the particles disperse more effectively.

The results of the TEM image analysis (shown in Figure 5) make the composition of the FAC composites quite evident. The Ag NPs and the Fe_3_O_4_ NPs are interconnected and dispersed across the surface of the CNC bearing. The amount of time spent in the hydrothermal process causes the particle size of the Fe_3_O_4_ to decrease gradually. Additionally, the particles become more densely packed and uniformly distributed.

Figure 6 shows the EDX spectra of the samples FA5 and FAC5. The EDX spectra of both the FA5 and FAC5 materials revealed the presence of O, Fe, and Ag peaks. In addition, the EDX spectrum of FAC5 shows a prominent signal peak at 0.27 keV, which is characteristic of the C element of the CNCs in the material.

The magnetic property of our materials was investigated using a vibrating sample magnetometer (VSM). All of our magnetic hysteresis is depicted in Figure 7. The saturation magnetization (M_s_), remanent magnetization (M_r_), and coercivity (H_c_) values of seven samples are displayed in Table 1 based on the results of the VSM; from Fe_3_O_4_ to FA to FAC, the Ms of the samples decreased. The saturation magnetization of Fe_3_O_4_ reduces when Ag nanoparticles are present on the Fe_3_O_4_ surface and reduces more with the presence of CNCs. This result is consistent with previous research [40,41,42].

In contrast to the FA materials, the Ms value of the FAC materials increases with increasing hydrothermal duration in the presence of CNCs. The phase composition of the materials may cause this result. As mentioned in the PXRD results, the FAC3 and FAC4 samples have low M_s_ due to impurity phases with diffraction peaks at 2*θ* = 27.7° and 32.1°. These peaks disappeared in the FAC5 sample after 5 h of hydrothermal treatment, so FAC5 has a higher Ms than the FAC3 and FAC4 samples. Even though the magnetism of FAC5 is significantly diminished compared to that of pure Fe_3_O_4_, it can be readily separated from the solution by applying an external magnetic field (Figure 7). This magnetic behavior not only makes the FAC5 catalyst economically viable for recovery and reuse but it also precludes the production of secondary sources of pollution, which are generated by catalyst residues that are not recovered after processing.

Figure 8 depicts the TGA and DTG of the CNCs and FAC5. According to Alvarez and Va1zquez, the decomposition temperature of cellulose is approximately 360 °C, where high-weight macromolecules are broken down into small-weight glucose units [43]. The DTG curves (Figure 8b) reveal that the CNCs have the highest decomposition temperature at 357.0 °C, and the initial decomposition temperature is relatively high at 315.0 °C. Upon hydrolysis by HCl acid, the hydroxyl groups on the cellulose surface tend to interact to form densely packed hydrogen bonding networks around nano cellulose, resulting in a higher decomposition temperature [44].

The TGA results (Figure 8a) indicate that after Ag and Fe_3_O_4_ are bonded to the CNC template, the material decomposes at approximately 284.0 °C and reaches its maximal temperature of 297.1 °C. It can explain that when the metal and metal oxide form on the CNC surface, the metal decomposes faster than the initial CNCs due to its high heat conductivity. In addition, a significant amount of FAC5 decomposes at 343 °C, with the highest decomposition temperature occurring at 364.6 °C. It is the thermal degradation region of cellulose, where the weight loss is relatively high, and the charred residue exceeds the initial CNCs (13.7%). According to FAC5, metals, and metal oxides have high heat resistance, and they remain in the structure after cellulose decomposes completely, resulting in the sample’s thermal stability. Therefore, FAC5 has a reduced decomposition rate and more char residue than the CNCs.

### 3.2. Catalytic Reduction of 4-NP

The reduction of 4-NP in excess NaBH_4_ was selected as a model reaction to evaluate the catalytic activity of the prepared FAC5 nanocomposite. UV–vis absorption spectroscopy was used to track the progression of the catalytic reduction. During the experiment, the peak UV absorption of an aqueous solution of 4-NP moved with increasing intensity from 318 to 400 nm after adding freshly generated NaBH_4_ solution. The production of 4-nitrophenolate ions with a more vital -conjugated donor-acceptor characteristic is probably responsible for the initial color change (mild to intense yellow) followed by a red shift when NaBH_4_ solution is added [45]. After a delay, the UV absorption peak of 4-NP at 400 nm, and its brilliant yellow color, disappeared upon adding the FAC5 catalyst. The development and progressive strengthening of a new peak at 300 nm confirms the progression of the reaction due to the production of 4-AP. Figure 9 depicts the absorption spectrum of 4-NP as a function of time for the catalysts FA5 and FAC5. In the presence of 2.5 mg FAC5, 4-NP was nearly completely reduced within 10 min, accompanied by a change in color from vibrant yellow to colorless. *C_t_* and *C*_0_ were the absorbance values of 4-NP in the presence of NaBH_4_ at time *t* = *t* and *t* = 0, respectively, and were plotted against time (*t*) in Figure 9 in the presence of the FA5 and FAC5 catalysts. FAC5 has higher catalytic activity than FA5. After 60 min, the absorption peak of 4-NP at 400 nm in NaBH_4_ and without the catalyst is almost unchanged.

Similarly, 4-NP reduction proceeded very slowly when it occurred in the presence of a catalyst but without NaBH_4_ as a reducer. These occurrences indicate that the reaction requires both the catalyst and a reducing agent (NaBH_4_). In addition, the potential catalytic activity of the CNCs and the Fe_3_O_4_ adsorbent was examined in the presence and absence of NaBH_4_ under conditions analogous to those stated for FAC5. After 60 min of reaction, however, the absorption peak intensity at 400 nm remained unchanged relative to that of the initial 4-NP, indicating that the Ag NPs played a crucial role in the conversion of -NO_2_.

### 3.3. Removal of MO and MB via Adsorption Process and Catalytic Reduction

At ambient temperature, the adsorption of two water-soluble organic dyes (MB and MO) was tested using CNC, FA5, and FAC5. In total, 15.0 mg of each substance was added to a 30 mL beaker containing a 1.0 × 10^−4^ M aqueous dye solution. The mixture was agitated at 300 rounds per min. The experiments were evaluated by measuring the absorbance of the dye solution at intervals and then determining the MB concentration using the dye calibration curve. The dye removal was calculated using the following formula:Dye removal = *C_t_*/*C*_0_ × 100(1)
where *C_t_* is the dye concentration at time *t*, and *C*_0_ is the initial dye concentration.

Figure 10a,c depict the (*C_t_*/*C*_0_) versus time profiles for determining the remaining MB and MO after using CNC, FA5, and FAC5 adsorbents. All materials generally possess MB adsorption, and the equivalence is reached in 10 min. Due to the electrostatic interaction between the negatively charged hydroxyl groups on the surface of the CNCs and the positively charged cation MB dye, the CNCs exhibit excellent MB adsorption. Twenty-five percent of the MB remained after a 30-min test on the CNCs. FA5’s adsorption MB is significantly lower than that of the CNCs. After 30 min of MB adsorption, the remaining MB concentration in FA5 was 48.3%. Upon the appearance of Ag/Fe_3_O_4_ on the CNC surface, the MB dye’s absorption increased dramatically compared to Ag/Fe_3_O_4_. The FAC5 material exhibited the highest adsorption efficacy, as shown in Figure 10a. After 30 min, the remaining concentration of MB was 10.5%. The presence of CNCs in the hydrothermal process facilitates the uniform dispersion of Ag/Fe_3_O_4_ on the CNC surface, increasing the surface area of FAC5 relative to FA5. The results shown in Figure 11a provide information regarding the two samples’ BET surface area (S_BET_). According to the International Union of Pure and Applied Chemistry (IUPAC) classification, the evaluated materials exhibited type IV patterns with hysteresis loops. Two samples exhibited narrow H3-type hysteresis loops (parallel plate-shaped apertures) based on the hysteresis loop patterns. In addition, the zeta potential values of the FAC5 suspension obtained through DLS analysis were –27.2 ± 1.1 mV negative (Figure 11b). This also facilitates enhanced cationic MB dye adsorption. MO, as opposed to MB, is a negative anion dye. Thus, CNCs hardly absorb MO. FA5 and FAC5 adsorption on MO also occurred and reached equilibrium after 10 min of agitating both substances in the MO solution. Due to its greater surface area, FAC5 has a higher MO adsorption capacity than FA5.

Given the outstanding adsorption behavior of the FAC5 nanocomposite for MB, our next objective was to investigate its performance in degrading hazardous organic dyes using NaBH_4_ as a reducing agent, and monitoring the variation in the intensity of the UV absorption peak at *λ*_max_ = 664 nm [46,47] allowed for an evaluation of the MB catalytic degradation process. Figure 10b also depicts the time-dependent change in the modification in the MB absorption spectrum. Within 10 min, the MB dye solution completely lost its color, from an intense blue color (MB) to colorless leucomethylene blue (LMB). In addition, the reduction of MO was comparable to that of MB. Figure 10d depicts the UV–vis absorption maximum variation at *λ*_max_ = 464 nm for MO [48,49] at various time intervals. As the reduction progressed, it was observed that the characteristic peak intensity decreased significantly and nearly disappeared within 10 min. There was no significant change in the distinct absorption peaks of MB and MO after 60 min of exposure without a catalyst or a reductant.

Figure 12 depicts a potential mechanism for reducing 4-NP and organic dyes using the FAC5 catalyst with NaBH_4_ based on the above experimental results. In addition, the experimental results indicate that the reduction rate is significantly accelerated in the presence of Ag metal on the catalyst’s surface compared to NaBH_4_ alone. Figure 12 shows that the hydrolysis of borohydride ions in an aqueous solution generates H_2_ gas and BO_2_^−^ [50,51,52]. As the hydrogen mediator for the reduction of 4-NP, 4-NP is deposited onto the surface of the Ag NPs to produce a silver hydride complex. The Ag NPs serve as redox catalysts for dye reduction by conveying electrons between donor species (BO_2_^−^) and acceptor molecules (MO or MB) [53,54,55,56]. The MB experiment observed that the colorless (reduced) form of MB (LMB) underwent sluggish aerial oxidation in an open atmosphere after 3–4 h. However, the characteristic blue color dissipated upon shaking, as the excess NaBH_4_ in the solution once again diminished it. Similar observations, called “clock” reactions, have been reported in the past [57]. In contrast, the reduced MO solution did not change color, even after several days, indicating no subsequent re-oxidation. As shown in Figure 12, the MO decolorization products were N, N-dimethyl-benzene-1,4-diamine, and 4-aminobenzenesulfonate, formed via hydrogenation and subsequent -NH-NH- bond dissociation [58].

The recyclability of the FAC5 nanocomposite was an essential factor for more cost-effective processes, and we examined the sample’s efficiency in repeated reaction cycles. The FAC5 nanocomposite could be readily separated from the solution by an external magnetic field due to its strong magnetic properties. The nanocomposite was utilized for the MB and 4-NP reduction after being washed five times with distilled water. After three repetitions of this procedure, the FAC5 nanocomposite remained stable and exhibited a high level of reactive activity, indicating its outstanding recyclability. As depicted in Figure 13, the FAC5 was effectively reused in three successive 10-minute cycles. It was observed that the catalyst had not significantly lost activity after three cycles. This result indicates that the FAC5 nanocomposite developed in the present study is exceptionally stable and resistant to multiple reuse cycles.

## 4. Conclusions

Using a straightforward protocol, a magnetic CNC nanocomposite decorated with Ag NPs (Ag/Fe_3_O_4_/CNC) was synthesized at ambient temperature without adding any reducing agent or stabilizer. In this green synthesis method, the CNC matrix was essential for reducing the Ag^+^ precursor and stabilizing newly formed Ag NPs on its surface. PXRD and EDX analyses confirmed the formation of Ag NPs on the CNC surface. TEM and FE-SEM investigations also confirmed that the magnetic nanocomposite’s surface was decorated with Ag NPs, and in the presence of NaBH_4_, the Ag/Fe_3_O_4_/CNC nanocomposite exhibited exceptional catalytic activity for reducing 4-NP to 4-AP. It appeared to be an impressive catalyst for the rapid reduction and decolorization of toxic organic substances (both MB and MO) in a few min (2.0–5.0) with a tiny amount of catalyst (2.5 mg for 4-NP and 15 mg for MB and MO). The hydrolysis of borohydride ions in an aqueous solution resulted in the formation of H_2_ gas and electron-rich BO_2_^−^ ions as intermediates for the construction of a hydrogen-mediator complex (silver hydride) and an electron relay system for the reduction of organic species (4-NP, MO, and MB). Therefore, Ag/Fe_3_O_4_/CNC is a potentially helpful catalyst for removing organic toxic pollutants from contaminated water bodies.

## Figures and Tables

**Figure 1 polymers-15-03373-f001:**
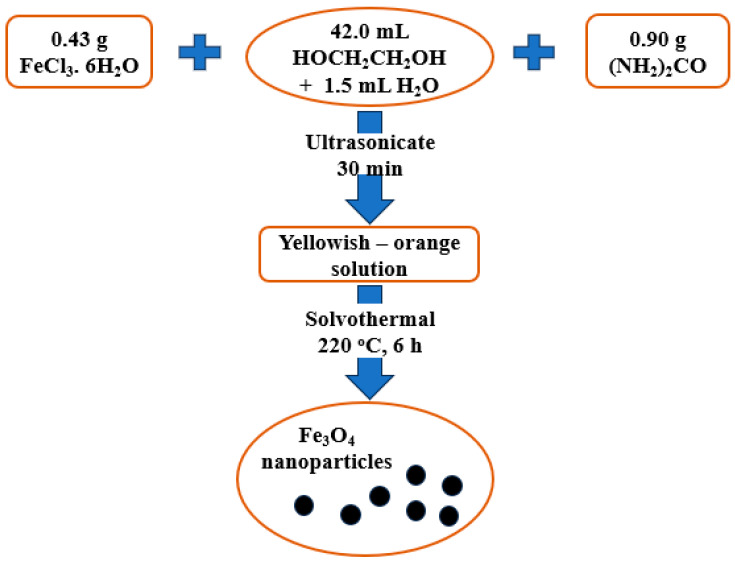
The synthesis process of Fe_3_O_4_ nanoparticles.

**Figure 2 polymers-15-03373-f002:**
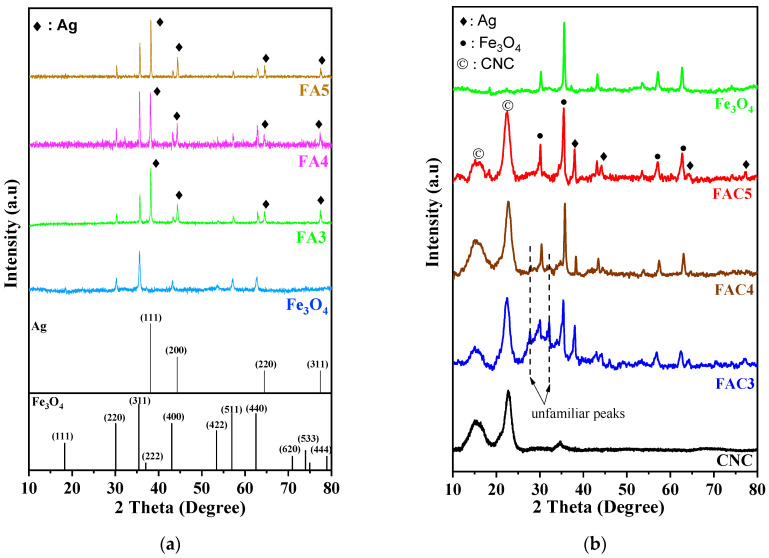
The PXRD patterns of (**a**) Fe_3_O_4_ and FA materials and (**b**) CNC and FAC materials.

**Figure 3 polymers-15-03373-f003:**
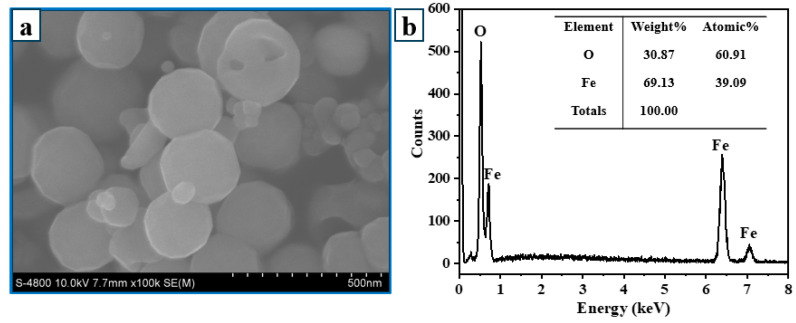
FE-SEM image (**a**) and EDX spectrum (**b**) of Fe_3_O_4_ NPs.

**Figure 4 polymers-15-03373-f004:**
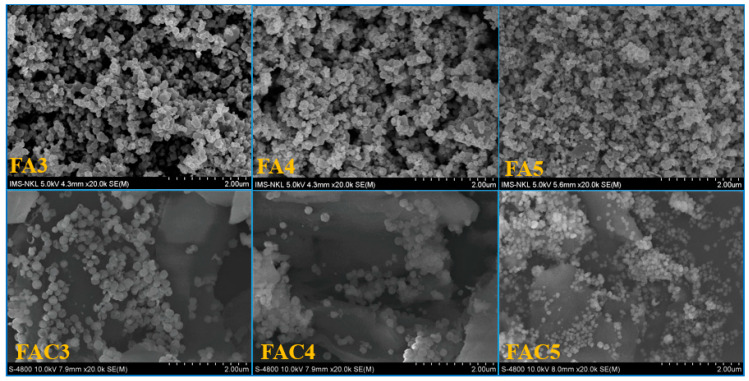
FE-SEM images of FA and FAC materials.

**Figure 5 polymers-15-03373-f005:**
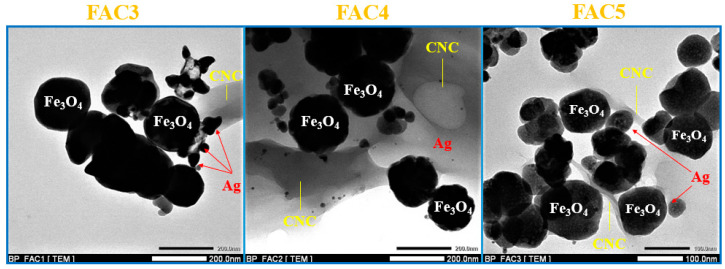
TEM images of FAC materials.

**Figure 6 polymers-15-03373-f006:**
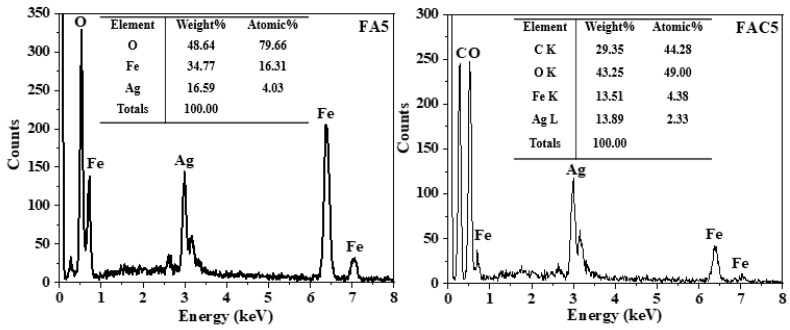
EDX spectra of FA5 and FAC5.

**Figure 7 polymers-15-03373-f007:**
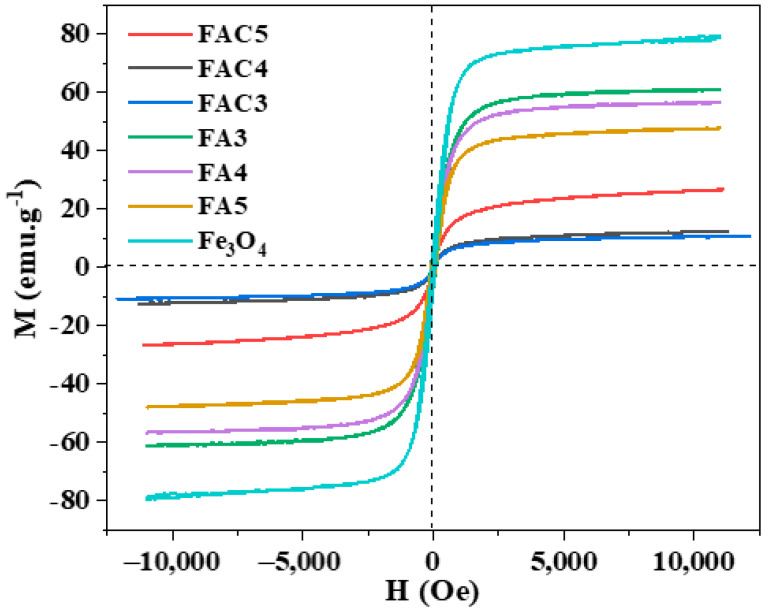
M-H loops of Fe_3_O_4_, FA, and FAC materials.

**Figure 8 polymers-15-03373-f008:**
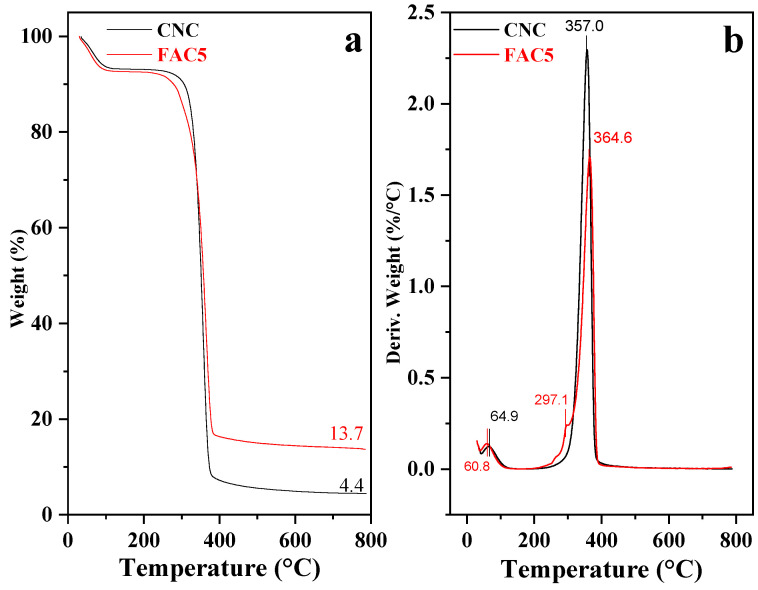
TGA (**a**) and DTG (**b**) curves of CNCs and FAC5.

**Figure 9 polymers-15-03373-f009:**
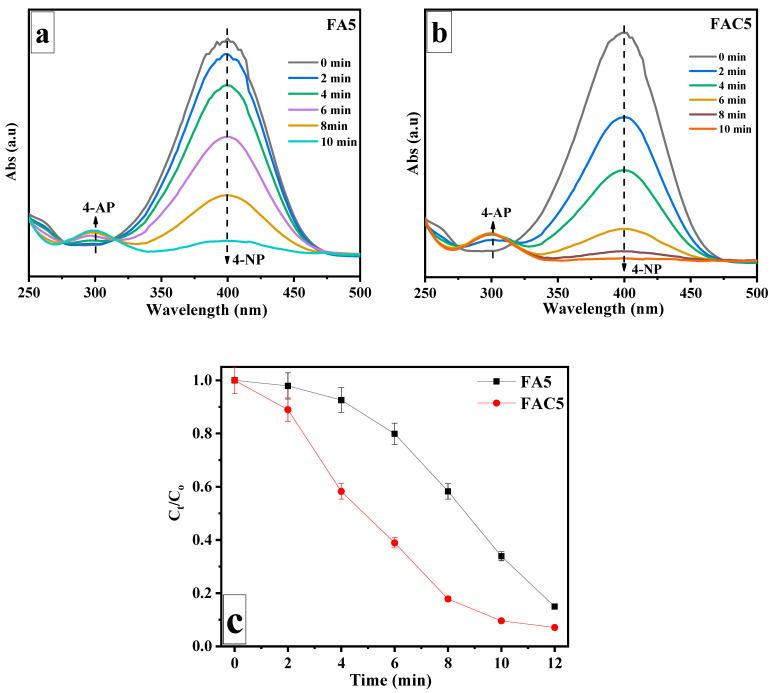
Changes in UV–vis absorption spectra for the reduction of 4-NP by NaBH_4_ over time in the presence of (**a**) FA5 and (**b**) FAC5 catalysts; (**c**) the corresponding plot of (*C_t_*/*C*_0_) versus reaction time (t).

**Figure 10 polymers-15-03373-f010:**
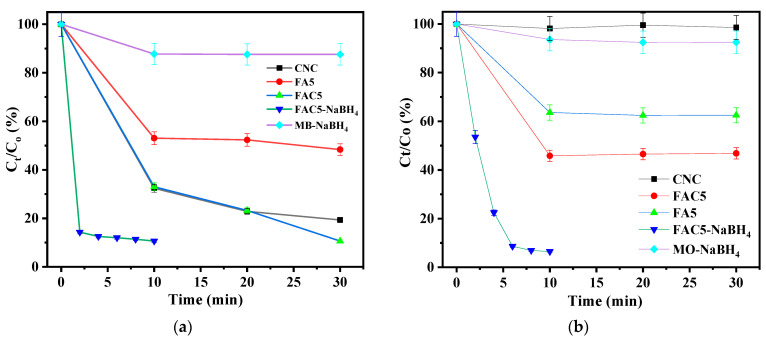
Graphs of (*C_t_*/*C*_0_) vs. time (t) for (**a**) MB and (**c**) MO; UV–vis absorption spectrum changes for the reduction of (**b**) MB and (**d**) MO by NaBH_4_ at different time intervals in the presence of FAC5 catalyst.

**Figure 11 polymers-15-03373-f011:**
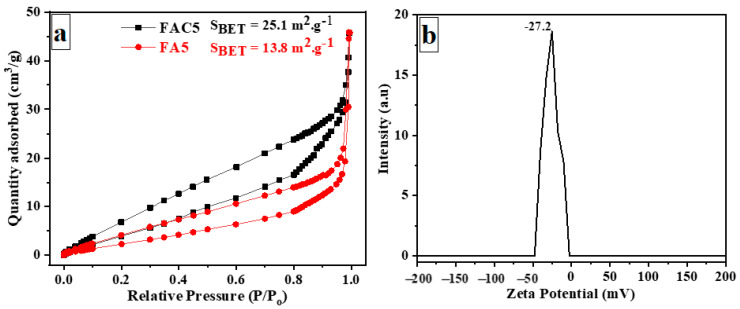
(**a**) Nitrogen adsorption and desorption isotherms of FA5 and FAC5 materials; (**b**) zeta potential distribution of FAC5 material.

**Figure 12 polymers-15-03373-f012:**
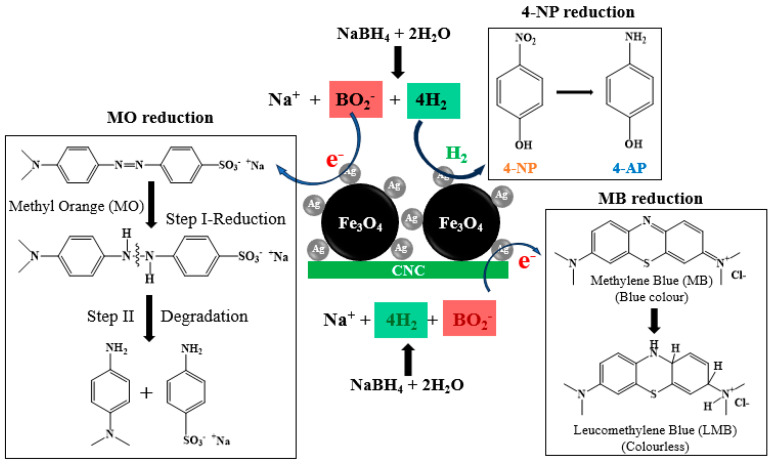
A possible way for 4-NP and organic dyes (MO and MB) to be broken down by FAC5 nanocomposite.

**Figure 13 polymers-15-03373-f013:**
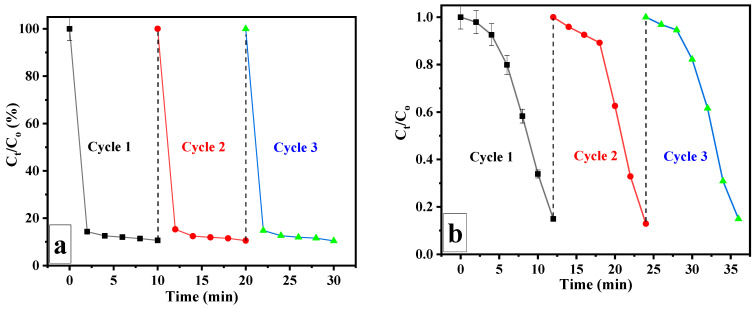
Reusability of FAC5 catalyst over three successive cycles for reduction of (**a**) MB and (**b**) 4-NP.

**Table 1 polymers-15-03373-t001:** The M_s_, M_r_, and H_c_ values of seven samples.

No.	Sample	M_s_ (emu/g)	M_r_ (emu/g)	H_c_ (Oe)
1	Fe_3_O_4_	78.5	6.5	61.6
2	FA3	61.1	7.7	102.6
3	FA4	56.4	5.9	75.2
4	FA5	47.7	5.9	82.1
5	FAC3	10.6	0.1	6.8
6	FAC4	12.4	0.2	15.9
7	FAC5	26.6	0.4	6.8

## Data Availability

The data presented in this study are available in this published article.

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
