# Peer review of "Facile Hydrothermal Synthesis of Ag/Fe3O4/Cellulose Nanocomposite as Highly Active Catalyst for 4-Nitrophenol and Organic Dye Reduction"

_polymers, 2023, doi:10.3390/polym15163373_

Round 1
Reviewer 1 Report
The work by Vu Nang An et al. entitled “FACILE HYDROTHERMAL SYNTHESIS OF Ag/Fe3O4/CELLULOSE NANOCRYSTALS NANOCOMPOSITE AS HIGHLY ACTIVE CATALYST FOR 4-NITROPHENOL AND TOXIC ORGANIC DYES” provides a solvothermal synthesis route for the production of Ag/Fe3O4/cellulose nanocatalysts for environmental remediation. This study is relevant to the field and would be a nice contribution once it has been revised. The length of the work is sufficient. I recommend this work for publication the following changes have been included.
The following comments should be considered when revising.
Lines 1-4: The title is wordy and a bit unclear. For example, either nanocrystals or nanocomposites could be used instead of both. Also, “ … active catalyst for 4-nitrophenol and toxic organic dyes” doesn’t sound very clear on the exact catalytic reaction. I’d like to suggest re-wording “ catalyst for 4-nitrophenol and organic dye reduction” or “catalyst for 4-nitrophenol and organic dye degradation”, etc.
Figure 1 has a few issues and is suggested to be revised as follows: ethylene glycol is abbreviated in the figure, but the abbreviation is not introduced anywhere in the manuscript. Urea is misspelled. Solvent thermal should be corrected as solvothermal. Including the specific amounts, time duration, and temperature conditions will enhance the clarity of the figure.
FA3, FA4, and FA5 should be introduced at some point in the synthesis (line 165 is suggested)
The term X-ray diffraction should be corrected as Powder X-ray diffraction (PXRD) to avoid confusion between powder and single crystal XRD techniques.
Line 173-174: The accelerating voltage of the SEM under which the imaging has been performed should be specified.
Line 215-216: “Figure 2. The XRD patterns of (a) Fe3O4 and FA materials synthesized at three different.
hydrothermal times”. Did the authors make more than one Fe3O4 sample by varying synthesis time durations? If not this figure caption should be re-worded. Also, here, Fe3O4 is mentioned hydrothermal whereas it was previously mentioned to be synthesized using the solvothermal method. This could be confusing to the reader.
Figure 4: The author compared FA and FAC in figure 4, however, their magnifications are quite different. The authors claim (lines 244-246) “In contrast to Fe3O4, the formation and attachment of Ag to the surface of Fe3O4 during the hydrothermal process gives FA spherical particles an uneven surface (Figure 4). The FAC material indicates that Fe3O4, Ag, or Ag/ Fe3O4 particles form on the CNC surface in the presence of the CNC.” not clear in Figure 4.
Figure 5: It’s not clear how the authors have identified/distinguished the Fe3O4, Ag, and CNC in the TEM images. In order to distinguish these particles in TEM micrographs, one must perform a spectroscopic study in the TEM/Z-contrast imaging, etc. Since the authors have identified and labeled these particles, the method of identification should be described.
Some sentences should be re-worded for clarity. For example Line 19-“Especially, Ag NPs were decorated on the CNC surface without reducing agents or stabilizers by reducing AgNO3.” This sentence is hard to understand, especially the “stabilizers by reducing AgNO3” part. I suggest rewording this as “The CNC surface was decorated with Ag NPs without using any reducing agents or stabilizers” to enhance the clarity. More examples: Lines 68-70, 79-80, 124-125, etc.
Reviewer 2 Report
Nang-An et al. submitted, "Facile Hydrothermal Synthesis Of Ag/Fe3o4/Cellulose Nanocrystals Nanocomposite As Highly Active Catalyst For 4-Nitrophenol And Toxic Organic Dyes" describe the synthesis of silver nanoparticle loaded magnetite onto the cellulose. The interdisciplinary study has a wide application in removing the dyes.
1. (a) Authors used words like "dyes" as well as "pigments" in some sections; For example, in line 46, the author stated, "azo-organic dye pigments," which gives a sense of natural pigments. However, the state of chemistry for these azo compounds (or colorants) is underestimated as there are a wide variety of organic dyes available at the commercial level, such as reactive dyes, dispersive, basic, or acidic-dyes, which must be clearly stated. if considering these as for coloring the materials, could be called dispersive dyes (while dispersive dyes generally tend not to have a charge on their surface) or otherwise an acidic dye (as 4-nitrophenol is a pH-based indicator used for titration reaction); please check carefully. However, the author is open to choosing accordingly what suits their interest.
(b) In my opinion, as the paper is a research article, the author needs to be careful in using broader terminologies. For example, the author used three compounds (para-nitrophenol, methyl orange, and methyl blue), which are said organic dyes (or sometimes pigments in the manuscript), are pH indicators, and generally, their dyeability (or sustain coloration properties) is average and better suited to say as "pH indicators". However, (by any chance), if we see them as a dye, exploiting their molecular charge (phenolate (OH) of nitrophenol) to interact with any cationic surface charged material (say example, cationic cellulose-based materials), which leads to enhance their surface adsorption (because of increased ionic-ionic interaction) onto the material, will eventually easily washed away (therefore, will have average washing fastness properties).
As a suggestion, the author could say, for example, Azo-colorants or azo-compounds rather than organic dyes. Please rewrite the title of the manuscript. However, these are just suggestions for authors, and authors can choose according to their interests.
2. The role of sodium borohydride NaBH4
The author stated that NaBH4 acts as an electron donor in the reaction (line 95 and line 96) to reduce the 4-nitrophenol, methyl Orange and methyl blue catalytically.
(a) Please explain the reaction briefly to provide an overview (if possible, use the chemical structures for explanation), as deciding the role of additives is confusing in the current form.
(b) Have authors performed any reaction where they omit the use of NaBH4 to understand the dependence of this reaction on NaBH4?
(c) If Na2BH4 acts as an electron donor, does the author try to use other electron donors (or reducing agents) which perform a similar role-like NaBH4.
3. Reaction Monitoring:
(a) Qualitative analysis: How was the reaction qualitatively analyzed for the formed products? Please state it clearly. In other words, how were the chemical structures of reaction products (reduced compounds) confirmed?
(b) Quantitative analysis:
(i) Is the catalytic reduction 100% achieved? if not, how much it was, and how it was calculated?
(ii) What instrumentation or methodology was used to quantify the yields of products formed? For example, researchers commonly use HPLC methods or NMR-based monitoring to calculate the yields. Please provide the information.
Sentence reformation is required.
Some sentences are written in complex ways; please break them into simple ones to improve the readability.
